# Insights into the Sulfate Transporter Gene Family and Its Expression Patterns in Durum Wheat Seedlings under Salinity

**DOI:** 10.3390/genes14020333

**Published:** 2023-01-27

**Authors:** Fatemeh Puresmaeli, Parviz Heidari, Shaneka Lawson

**Affiliations:** 1Faculty of Agriculture, Shahrood University of Technology, Shahrood 3619995161, Iran; 2USDA Forest Service, Northern Research Station, Hardwood Tree Improvement and Regeneration Center (HTIRC), Department of Forestry and Natural Resources, Purdue University, 715 West State Street, West Lafayette, IN 47906, USA

**Keywords:** plant gene families, nutrient transporters, abiotic stresses, sulfur, gene expression

## Abstract

Sulfate transporters (SULTRs) are an essential plant transporter class responsible for the absorption and distribution of sulfur, an essential plant growth element. SULTRs are also involved in processes related to growth and development and in response to environmental stimuli. In the present study, 22 *Td*SULTR family members were identified and characterized in the genome of *Triticum turgidum* L. ssp. *durum* (Desf.) using available bioinformatics tools. The expression levels of candidate *TdSULTR* genes were investigated under salt treatments of 150 and 250 mM NaCl after several different exposure times. *Td*SULTRs showed diversity in terms of physiochemical properties, gene structure, and pocket sites. *Td*SULTRs and their orthologues were classified into the known five main plant groups of highly diverse subfamilies. In addition, it was noted that segmental duplication events could lengthen *Td*SULTR family members under evolutionary processes. Based on pocket site analysis, the amino acids leucine (L), valine (V), and serine (S) were most often detected in *Td*SULTR protein binding sites. Moreover, it was predicted that *Td*SULTRs have a high potential to be targeted by phosphorylation modifications. According to promoter site analysis, the plant bioregulators ABA and MeJA were predicted to affect *TdSULTR* expression patterns. Real-time PCR analysis revealed *TdSULTR* genes are differentially expressed at 150 mM NaCl but show similar expression in response to 250 mM NaCl. *TdSULTR* reached a maximum level of expression 72 h after the 250 mM salt treatment. Overall, we conclude that *TdSULTR* genes are involved in the response to salinity in durum wheat. However, additional studies of functionality are needed to determine their precise function and linked-interaction pathways.

## 1. Introduction

Sulfur, an important and necessary element for optimal plant growth and development, is involved in many cellular processes [1]. In addition, sulfur is present in the structure of hormones, vitamins, amino acids, and coenzymes, and its deficiency will reduce the quantity and quality of plant production [2]. Sulfur is also involved in the formation of glutathione, which is sensitive to oxidative stress and functions to regulate oxidant-dependent signaling pathways along with stress-related responses [1,3,4,5]. Also, sulfur-dependent metabolites are effective at increasing the tolerance of plants to abiotic stresses such as salt stress by controlling and regulating related molecular and physiological processes. Plants absorb sulfate through root cells and use it for sulfur-dependent metabolism. Absorption and distribution of sulfate occur through sulfate transporters (SULTRs) located in cell membranes and those of organelles such as the vacuole and plastid [6,7]. SULTRs contain a STAS (Sulfate Transporter/AntiSigma-factor) domain in its carboxyl-terminal region and 12 membrane-spanning domains [8,9,10].

Due to the importance of sulfur to subsequent plant growth processes, the SULTR gene family has been subjected to detailed molecular investigation in model plants such as *Arabidopsis* [11,12,13,14,15]. Moreover, the SULTR gene family is classified into five groups and four putative subfamilies, including subfamily SULTR 1, 2, 3, and 4 [11,12,13,14,15]. The function of some members of these subfamilies has been determined. For example, subfamily SULTR 1 members, 1.1, 1.2, and 1.3, are primarily active in roots and are responsible for sulfate absorption and transport [11,12]. SULTR 2.1 and SULTR 2.2 belong to group 2, which play principal roles in transferring sulfate from roots to other plant organs [14]. Functional activities for the five members of the SULTR 3 subfamily (SULTR 3.1, 3.2, 3.3, 3.4, and 3.5) vary and a specific function has not been determined [16]. SULTR 4.1 and SULTR 4.2, members of group 4, are involved in the transport and transfer of sulfate between the vacuole and the cytosol [6,17]. SULTRs are predominantly involved in growth and development processes, but studies have revealed that members of this gene family are also involved in responding to environmental stress [18,19,20,21]. For example, subfamily SULTR 3 members have been found to interact with transcription factors (TFs) associated with stress responses in potatoes [19]. It was reported that *SULTR* genes in maize are induced as a response to heat and drought stress [22]. Heavy metal stress will also alter SULTR expression [20,21].

*Triticum turgidum* L. ssp. *durum* (Desf.) is a tetraploid wheat, 2 n = 4 × = 28, that is well adapted to arid regions [23,24]. As mentioned above, SULTRs play critical roles in the absorption and transfer of sulfate and in responses to environmental stimuli. Some SULTR gene family members have been identified and characterized in several plant models and the functionality of some gene family members has been determined. However, this gene family has not been studied in *T. turgidum* L. ssp. *durum* (Desf.). The SULTR family genes of *T. turgidum* L. ssp. *durum* (Desf.) (*Td*SULTRs) studied here were identified and characterized with the expression patterns of candidate *TdSULTRs* evaluated under salinity.

## 2. Materials and Methods

### 2.1. Identification of SULTRs

To recognize the SULTR family members in the genome of *T. turgidum* L. ssp. *durum* (Desf.) (*Td*SULTRs), two domains Sulfate_transp (PF00916), and STAS (PF01740) were applied as queries with the BLASTP tool of Ensembl Plants database (Accessed: 20 December 2022) [25]. SULTR proteins were also obtained for the *Oryza sativa* Indica group, *Triticum urartu, Triticum aestivum*, *Hordeum vulgare*, *Sorghum bicolor*, and *Zea mays*. The presence of SULTR domains in identified proteins was investigated using the Pfam database (Accessed: 20 December 2022) [26] and the Conserved Domain Database (CDD) (Accessed: 20 December 2022) [27]. Several physiochemical properties of *Td*SULTRs, such as isoelectric points (pI), molecular weight (MW), GRAVY, and instability index were evaluated by the ExpasyProtParam tool (Accessed: 20 December 2022) [28]. The subcellular localization of each *Td*SULTR was predicted using the BUSCA server [29].

### 2.2. Phylogenetics and Conserved Motif Analyses

To construct a phylogeny tree, the SULTR proteins from *T. turgidum* L. ssp. *durum* (Desf.), *O. sativa*, *T. urartu*, *T. aestivum*, *H. vulgare*, *S. bicolor*, and *Z. mays* were analyzed by Clustal-Omega, an online multiple alignment tool [30]. Next, the aligned sequences were used to construct a phylogeny tree using the Maximum likelihood (ML) with 1000 bootstrap replication and the default setting with IQ-TREE web server [31]. The resultant file was imported into the iTOL online tool [32] to illustrate the phylogeny tree.

### 2.3. Duplication Analysis of TdSULTR Genes

To identify duplicated *TdSULTR* genes, the coding sequence of pairs of *TdSULTR* genes was compared and screened for genes with an identity ≥0.85 [33]. Furthermore, the non-synonymous (Ka) and synonymous (Ks) indexes were calculated for each duplicated gene by MEGAX software [34]. In addition, the time of divergence for duplicated genes was calculated using the following equation, T = (Ks/2λ) × 10^−6^. (λ = 6.5 × 10^−9^).

### 2.4. Promoter Analysis

To identify putative *cis*-regulatory elements in the promoter regions of *TdSULTR* genes, the Plant CARE tool (Accessed: 14 December 2022) [35] screened the 1500 bp region upstream of each gene. All discovered *cis*-regulatory elements were separated by function.

### 2.5. Prediction of Three-Dimensional Structure of TdSULTR Proteins

In the current study, the Phyre2 [36] database (Accessed: 17 December 2022) was used to predict the 3D structure of *Td*SULTR proteins. Predictions of pocket site locations as ligand binding regions in the 3D structure of *Td*SULTR proteins were generated with the Phyre investigator tool from the Phyre2 server (Accessed: 17 December 2022).

### 2.6. Prediction of Phosphorylation Region in TdSULTR Proteins

The region of phosphorylation events based on three amino acids, serine, tyrosine, and threonine, was predicted in *Td*SULTR proteins using the NetPhos 3-1 server (Accessed: 18 December 2022) [37], and the potential value of each region was set with a high probability, more than 0.90.

### 2.7. Interaction Network of SULTR Family

Protein-protein interaction of *Td*SULTRs was investigated with the String database (Accessed: 13 December 2022) based on orthologues from the model plant *Arabidopsis* [38]. Direct and indirect interactions were discerned by setting the first layer of the network to 5 and the second layer to 20 nodes. In addition, significant gene ontology (GO) terms (FDR ≤0.01) were identified based on the molecular function and biological processes of the interaction network nodes. The final interaction network was designed using Cytoscape software (v3.9.1) [39].

### 2.8. Plant Materials and Treatments

The present study utilized the *T. turgidum* L. ssp. *durum* (Desf.) cultivar Yavaros. Sterilized seeds were planted three per pot in peat moss. Seedlings were grown under photoperiodic lighting (16 h of light: 8 h of dark) at a temperature of 24 ± 3 °C. Six-week-old seedlings were separately treated with two salt concentrations, 150 and 250 mM NaCl, by irrigation. The process was repeated after 24 h with three additional pots irrigated without salt for use as controls. In the next step, seedling shoots were collected after 6, 24 and 72 h of salt treatment. The harvested shoot samples were immediately frozen in liquid nitrogen before being transferred to a −65 °C freezer.

### 2.9. Real-Time PCR Analysis

The total RNA of collected samples was extracted using an RNX plus kit (Cinaclone, Tehran, Iran), following the included manufacturer’s protocol. Also, a reverse transcriptase (Roche, Mannheim, Germany) enzyme was used to synthesize our cDNA based on the provided manufacturer’s protocol. In this study, six *TdSULTR* genes were selected for expression pattern analysis in response to salinity. The *Actin7* gene was selected as a housekeeping gene for data normalization. Primers (forward and reverse) for each selected gene (Appendix A) were designed using an online tool, Primer3 (Accessed: 10 December 2022) [40]. The observed expression patterns for the *TdSULTR* genes were evaluated by ABI Step One using Maxima SYBR Green/ROX qPCR Master Mix (Thermo Fisher, Illkirch-Graffenstaden, France), according to the manufacturer’s protocol. Lastly, gene expression levels were estimated using the delta-delta ct protocol [41]. Three biological replicates were used in this experiment.

## 3. Results

### 3.1. Identification and Physicochemical Properties of the SULTR Family

Twenty-two *TdSULTR* genes were identified within the *T. turgidum* L. ssp. *durum* (Desf.) genome and their physicochemical properties are provided in Table 1. *Td*SULTR proteins ranged in length from 147 (TRITD6Av1G100790) to 698 (TRITD4Bv1G165010) amino acids. Molecular weight (MW) ranged from 16.77 (TRITD6Av1G100790) to 75.69 kDa (TRITD4Bv1G165010). In addition, exon numbers varied from 3 to 13. The endomembrane system and organelle membranes were predicted to be the subcellular localization site for *Td*SULTR (Table 1). Based on isoelectric point (pI) data, most members of the *Td*SULTR gene family, with the exception of two proteins (TRITD6Av1G100790 and TRITD7Av1G260730), were alkaline (pI ≥ 7.0) in nature. The positive value for GRAVY indicates the hydrophobic nature of the proteins, while negative values indicate the hydrophilic nature of the proteins [42]. The value of GRAVY of all *Td*SULTR family members was positive, which indicates the hydrophilic nature of *Td*SULTRs. Results showed that *Td*SULTR family members have diverse physiochemical and structural properties, suggesting the SULTR gene family has likely been influenced by evolutionary processes.

### 3.2. Phylogenetic Analysis of SULTR Family Members

A phylogenetic tree of 22 *Td*SULTR proteins and associated orthologues in *T. aestivum*, *T. urartu*, *S. bicolor*, *H. vulgare*, *O. sativa*, *A. thaliana*, and *Z. mays* was constructed based on the maximum likelihood (ML) method (Figure 1). The results indicated SULTR family proteins could be classified into five main groups. The greatest diversity of SULTR proteins was observed in group IV. In addition, the highest number of *Td*SULTR proteins was observed in group III. Group IV was subdivided into two large subgroups, IV-a and IV-b. Group V was the largest clade with 32 SULTR proteins. Groups I, II, III and IV included 12, 22, 27, and 29 SULTR proteins, respectively. All SULTR 3.5 proteins were located in group I, while SULTR 3.1 and 3.2 were placed in group II. Other SULTR 3 proteins, SULTR 3.3 and 3.4, were located in group III, while all SULTR 1 proteins were in group V. Interestingly, SULTR 4 subfamily members along with SULTR 2 members were placed in group IV. A perusal of the phylogeny tree indicated the diversity in SULTR gene family members likely occurred after the derivation of monocots and dicots.

### 3.3. Chromosome Location and Duplication Events in TdSULTR Genes

Chromosome positioning for the 22 *TdSULTR* genes of *T. turgidum* L. ssp. *durum* (Desf.) illustrated that all were located on 12 chromosomes, an indication that *TdSULTR* genes were randomly distributed among the chromosomes (Figure 2a). For instance, four genes in chromosome 4A, three chromosomes 4B and 5A, two genes in chromosomes 5B and 7A, and a single gene in the 2A, 2B, 3A, 3B, 6A, 6B, and 7B chromosomes. Based on duplication analysis, 20 genes were segmentally duplicated among *Td*SULTR gene family members (Appendix A). The first duplication event was estimated to have been around 32 million years ago (MYB) for two *SULTR3.3* genes including *TRITD2Bv1G237240* and *TRITD7Av1G260730* (Figure 2b). The Ka/Ks ratio was calculated for duplicated genes (Figure 2c). All duplicated *TdSULTRs* showed Ka/Ks < 0.40, indicating that the duplicated *TdSULTR* genes were under purifying (negative) selection.

### 3.4. Analysis of TdSULTR Protein Structure

SULTR protein structures were modeled with more than 90% confidence and their ligand binding sites (pocket sites) were also predicted (Figure 3). According to the 3D structure results, different binding sites were predicted in *Td*SULTR proteins (Figure 3a). Variation in protein structure may reflect their transport activity in response to environmental stimuli. Based on pocket site analysis, leucine (L), valine (V), and serine (S) were frequently observed in the binding residues at the ligand binding site for nearly all of the *Td*SULTR proteins (Figure 3b). L was present in most *Td*SULTR proteins, while V and S were absent in three *Td*SULTR proteins each. Generally, L is known as a key residue in predicted pocket sites of *Td*SULTR proteins.

### 3.5. TdSULTR Post-Translational Modifications and Predicted Phosphorylation Regions

Potential phosphorylation regions, important for post-translational modifications, were investigated in *Td*SULTR proteins. Results revealed *Td*SULTRs are extremely likely (>90%) to target kinases (Figure 4). *Td*SULTR 3 subfamily protein TRITD4Bv1G165010, with 67, possessed the highest number of potential sites. It was also uncovered that serine is more influenced by phosphorylation modifications than the amino acids tyrosine and threonine.

### 3.6. Promoter Analysis of TdSULRs

Analysis of *TdSULTR* gene promoter regions led to the discovery of the putative cis-regulatory elements affecting expression patterns (Appendix A and Figure 5). In the present study, *cis*-regulatory elements were classified into four main groups, including hormone-responsive elements (REs), growth REs, stress REs, and light REs (Figure 5a). *Cis*-regulatory elements involved in plant stress responses were most abundant, at 35%, with *cis*-regulatory elements involved in growth processes (8%) those least represented (Figure 5a). *Cis*-regulatory elements related to hormone responses were divided into five groups, including auxin REs, methyl jasmonate (MeJA) REs, abscisic acid (ABA) REs, GA Res, and salicylic acid (SA) REs (Figure 5b). ABA REs and MeJA REs were distributed most in the promoter region of *TdSULTRs*, indicating these gene family members are more induced by ABA and MeJA. In addition, *cis*-regulatory elements involved in response to drought, anaerobic, and low-temperature stresses are most often found upstream sites of *TdSULTR* genes (Figure 5c).

### 3.7. Expression Profiles of TdSULTR Genes

The expression patterns of members of a gene family can reveal some aspects of their function in response to different biological conditions. To clarify the potential biological roles of durum wheat *TdSULTR* genes, the expression patterns of six candidates were investigated under two levels of salt stress (150 and 250 mM NaCl concentration). Candidate genes showed differential expression patterns in response to salinity (Figure 6). The *TRITD3Bv1G170350* gene, from the *Td*SULTR 3.5 subfamily, showed a significant down-regulation after 6 h of salinity in both applied salt concentrations, indicating that this gene can be an early response of durum wheat in response to salinity. Moreover, *TRITD3Bv1G170350* illustrated down-regulation in all series after 250 mM of NaCl. Besides, the *TRID4Av1G015890* gene (*Td*SULTR 1.2 subfamily) was also down-regulated in all time periods of 150 mM salt treatment, but up-regulated at 250 mM NaCl. *TRID5Bv1G12400*, a *Td*SULTR 4 subfamily member, was induced 24h after salt treatment and showed high up-regulation after 72 h of 250 mM NaCl. Thus, the *TRID5Bv1G12400* gene remains inactive during the early salinity stress response. *TRID7Av1G260730* (*Td*SULTR 3.3 subfamily), showed differential expression at 150 µM NaCl, up-regulation after 6h, and down-regulation after 24 h and 72 h of salt treatment. Two *Td*SULTRs, *TRID4Bv1G165010* (*Td*SULTR 3.1 subfamily) and *TRID4Bv1G157680* (*Td*SULTR subfamily 2), showed similar expression patterns with both genes showing upregulation at each time point, except 6 h after 150 mM NaCl. Interestingly, all candidate genes, with the exception of *TRITD3Bv1G170350,* displayed the highest expression levels 72 h after 250 mM salt treatment.

### 3.8. TdSULTR Interaction Network

In this study, the SULTR protein interaction network revealed proteins associated with sulfur assimilation such as SERAT, serine O-acetyltransferase, APS (adenosine 5′-phosphosulfate), APR (adenosine 5′-phosphosulfate reductase), and APK (adenosine-5′-phosphosulfate kinase) (Figure 7a). Gene ontology enrichment analysis based on network elements revealed that molecular function terms including adenylyl sulfate transferase (ATP) activity, adenylyl sulfate kinase activity, and adenylyl sulfate reductase (glutathione) activity were significantly linked with the SULTR network (Figure 7b). Biological processes such as the hydrogen sulfide biosynthesis process and sulfate reduction process were also significantly associated with the SULTR network. Cellular component enrichment analysis showed chloroplast and chloroplast stroma were the target sites of members of the SULTR network. As expected, based on KEGG analysis, sulfur metabolism was recognized as a metabolic process related to the SULTR network.

## 4. Discussion

Sulfur is a nutrient factor is involved in photosynthesis and critical growth processes and in plant responses to environmental stresses. Therefore, sulfur absorption and distribution in plant organs are very important. The sulfate transporters described here, as the main factor of sulfate distribution, help to improve plant performance of [18,43]. In the present study, 22 *Td*SULTR family members were identified and characterized by bioinformatic analyses. Physicochemical property analysis disclosed that *Td*SULTR were hydrophilic proteins, indicating that all *Td*SULTRs probably work under the same conditions. However, *TdSULTRs* were diverse based on exon number, ranging from 3 to 13, and more exons were observed in the structure of subfamily SULTR 3 genes. Exon numbers can affect gene expression thus, genes with high exon numbers are slowly induced compared to genes with fewer exons [44]. The phylogeny analysis showed the evolutionary process of each *Td*SULTR subfamily varied. In comparisons of relationships between the subfamilies of monocot SULTRs with the dicot model, *Arabidopsis*, it can be concluded that diversity within this family most likely occurred after the derivation of monocots and dicots [45,46]. In addition, subfamily *Td*SULTR 3 showed more diversity, suggesting that this subfamily may be an ancestor of other subfamilies [47,48]. Segmental duplication was recognized as a main evolution event, which expanded the number of *Td*SULTR family members. However, it was predicted that duplicated *TdSULTR* were under purifying selection based on Ka/Ks ratio [49]. Predicting three-dimensional structures of *Td*SULTR proteins showed that members of this gene family are structured similarly but possess different pocket regions. Variations in protein active site location can influence interactions and levels of protein activity [50,51]. L, V, and S were most abundant in the pocket regions, which showed the importance of these amino acids to *Td*SULTR activity rates and possible interactions. Hence, additional research is needed to better determine the effect of divergence in these areas. The *Td*SULTRs were predicted to be proteins highly likely to be phosphorylated by kinases. The presence of high-potential phosphorylation regions indicates *Td*SULTRs may be controlled by kinase-dependent signaling pathways. Studying expression patterns of target genes can provide a relative understanding of their biological functions. Few studies have been conducted in the field that addresses the function of SULTRs [6,7,52,53] and their roles in the plant environmental stress response are unknown. In this work, expression profiles of candidate *TdSULTRs* were investigated under two different concentrations of salinity, 150 and 250 mM NaCl. *TdSULTRs* showed diverse expression patterns in response to 150 mM NaCl, while all *Td*SULTRs, except *TRITD3Bv1G170350,* were highly induced by 250 mM of NaCl after 72h. These results indicate that the expression pattern of *SULTRs* is dependent on salt concentration and duration of stress. Also, it seems that *TdSULTRs* are present in both early and delayed responses to salt stress. For instance, *TRID5Bv1G12400*, as a member of subfamily *Td*SULTR 4, could not be classified in the group of early responses to salinity. Hormones, cytosol Ca concentration, and kinases as well as ROS greatly induced plant cell signaling elements in response to stresses [54]. Cis-regulatory elements related to ABA and MeJA-responsive hormones were often found upstream of *TdSULTR* genes, indicating *TdSULTR* genes can be controlled by stress-related hormones. These results indicate *sulfate transfer* genes are diverse in structure and function and are involved in several different cellular pathways.

## 5. Conclusions

In the present study, the structure and function of sulfate transporter gene family members in *T. turgidum* L. ssp. *durum* (Desf.) (*Td*SULTRs) were investigated. We conclude that *Td*SULTRs are diverse based on their physiochemical properties and structure. It also appears that *TdSULTRs* were extended by segmental duplication events. Moreover, we found that *TdSULTR* genes are involved in the hormone response to salinity and have the potential to induce responses from phytohormones such as ABA and MeJA. However, further functional analyses are required to understand the role of *TdSULTRs* in durum wheat resistance to stress and to find the upstream elements induced by the *TdSULTRs.*

## Figures and Tables

**Figure 1 genes-14-00333-f001:**
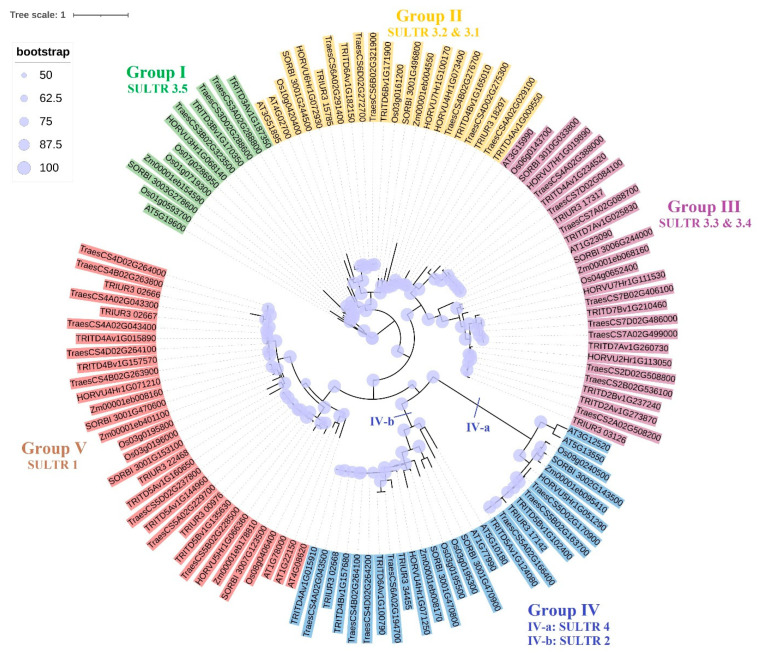
Phylogenetic analysis of SULTR gene family members from *T. turgidum* L. ssp. *durum* (Desf.) (TRITD prefix), *T. aestivum* (Traes prefix), *T. uratu* (TRIUR prefix), *S. bicolor* (SORBI prefix), *H. vulgare* (HORVU prefix), *O. sativa* (Os prefix), *A. thaliana* (AT prefix), and *Z. mays* (Zm prefix).

**Figure 2 genes-14-00333-f002:**
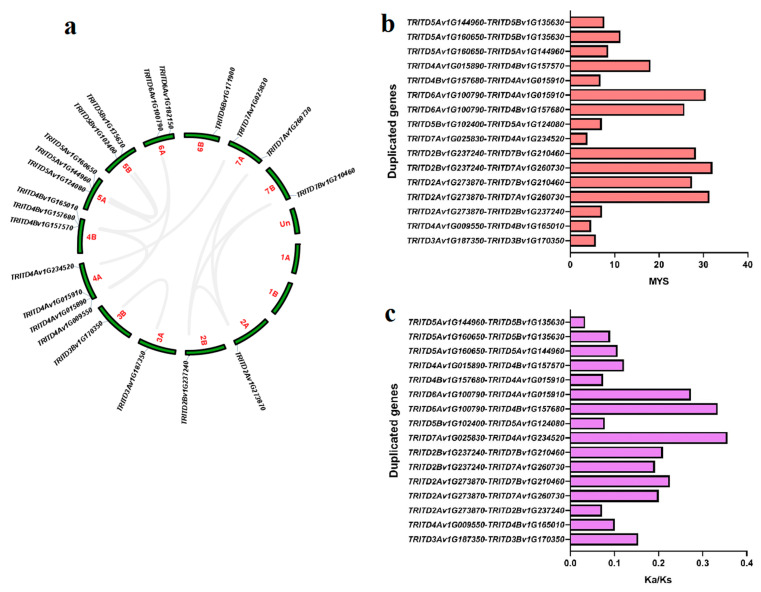
Distribution of *TdSULTR* genes in the genome of *T. turgidum* L. ssp. *durum* (Desf.)*,* gray lines showed the duplicated genes (**a**), time of divergence of duplicated *TdSULTR* genes based on million years ago (**b**), and the ratio Ka/Ks of duplicated genes of the *Td*SULTR gene family (**c**).

**Figure 3 genes-14-00333-f003:**
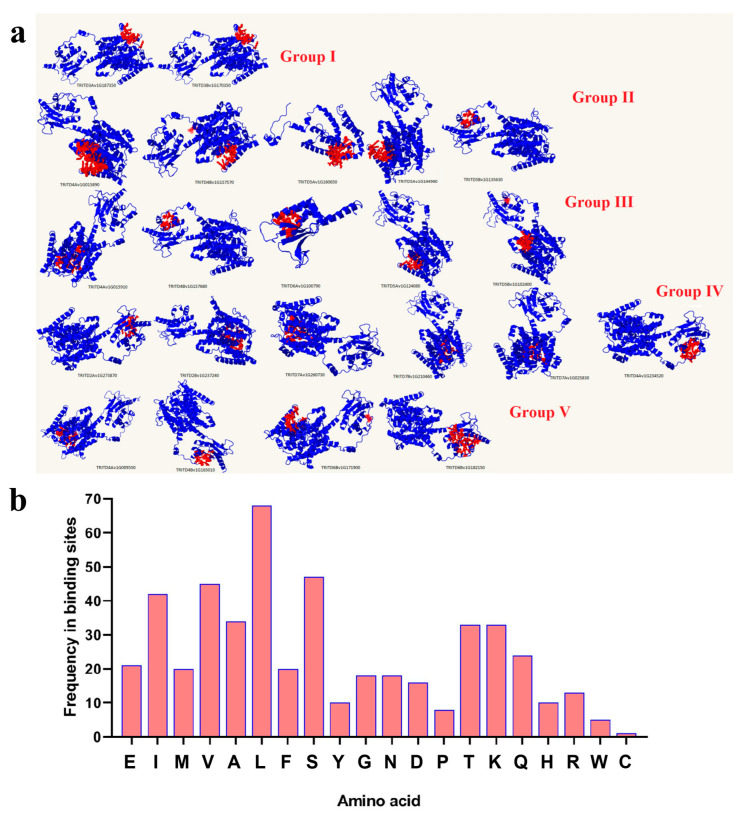
Three-dimensional and pocket site analysis of *Td*SULTRs in *T. turgidum* (**a**), and frequency of residues located in pocket sites of *Td*SULTRs (**b**). Red fill indicates the pocket site in the 3D structure of *Td*SULTRs. Abbreviations: Alanine, (Ala, A); Arginine, (Arg, R); Asparagine, (Asn, N); Aspartic acid, (Asp, D); Cysteine, (Cys, C); Glutamic acid, (Glu, E); Glutamine, (Gln, Q); Glycine, (Gly, G); Histidine, (His, H); Isoleucine, (Ile, I); Leucine, (Leu, L); Lysine, (Lys, K); Methionine, (Met, M); Phenylalanine, (Phe, F); Proline, (Pro, P); Serine, (Ser, S); Threonine, (Thr, T); Tryptophan, (Trp, W); Tyrosine, (Tyr, Y); Valine, (Val, V).

**Figure 4 genes-14-00333-f004:**
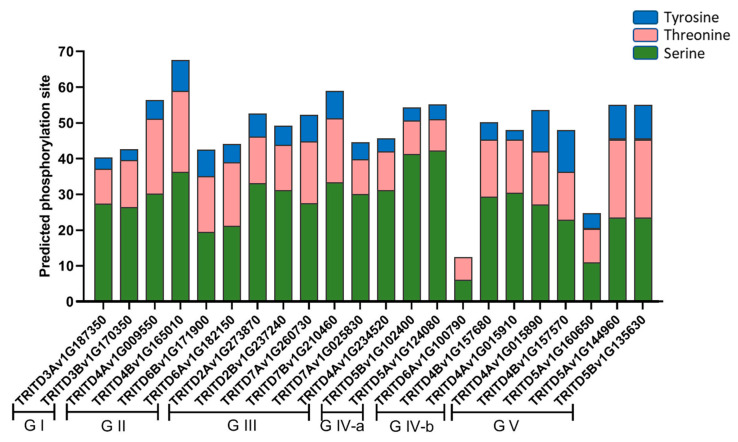
Prediction of phosphorylation sites in *Td*SULTR proteins based on three amino acids including serine, tyrosine, and threonine. *Td*SULTR is separated according to the phylogeny tree from G I (group 1) to G V (group 5).

**Figure 5 genes-14-00333-f005:**
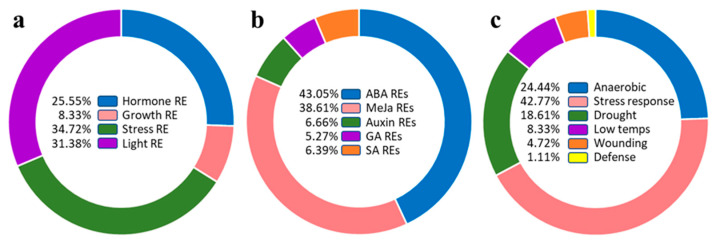
Classification of putative *cis*-regulatory elements present in the upstream region of *TdSULTR* genes (**a**), involved in response to phytohormones (**b**), and stresses (**c**). Full details are provided in Appendix A.

**Figure 6 genes-14-00333-f006:**
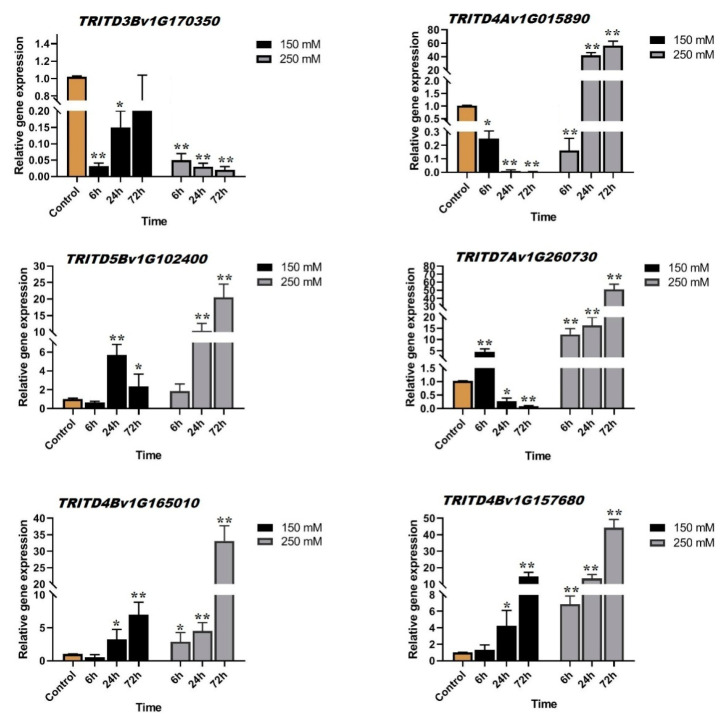
Expression levels of candidate *TdSULTR* genes in durum wheat in response to salinity, 150 and 250 mM of NaCl. * and ** indicate a significant difference between the experimental treatments and control treatment (according to Student’s t-test) at *p* < 0.05 and *p* < 0.01, respectively.

**Figure 7 genes-14-00333-f007:**
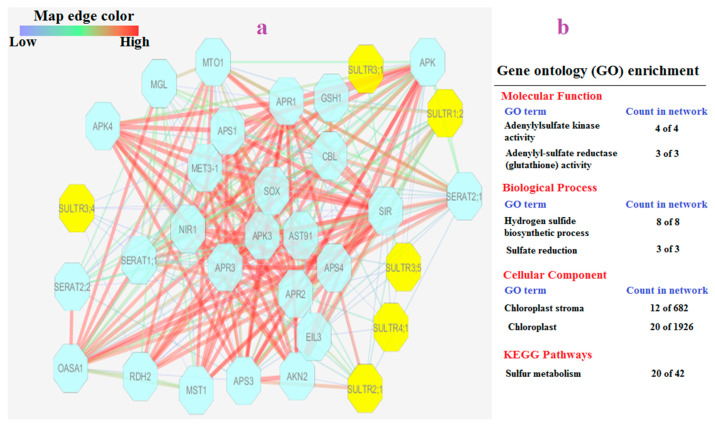
Interaction network of SULTR family members (**a**), and list of significant GO terms (FDR < 0.01) of nodes presented in network (**b**). Abbreviations: SERAT (serine O-acetyltransferase); APS (adenosine 5′-phosphosulfate); APR (adenosine 5′-phosphosulfate reductase); and APK (adenosine-5′-phosphosulfate kinase).

**Table 1 genes-14-00333-t001:** Physicochemical properties for *Td*SULTR family members within the *T. turgidum* L. ssp. *durum* (Desf.) genome.

Gene ID	Stability	MW (KDa)	pI	GRAVY	CDS (bp)	Exon Number	Peptide (aa)	Subcellular
TRITD6Av1G100790	Stable	16.77	6.12	0.228	444	5	147	Localization
TRITD4Av1G015890	Stable	72.88	9.05	0.326	1995	12	664	Organelle
TRITD4Bv1G157570	Stable	72.8	9.17	0.307	1989	10	662	membrane
TRITD7Av1G260730	Stable	72.06	6.68	0.43	1983	13	660	Endomembrane
TRITD7Bv1G210460	Stable	72.23	8.86	0.465	1983	13	660	Endomembrane
TRITD4Av1G234520	Stable	71.48	9.4	0.541	2004	7	667	-
TRITD4Av1G009550	Stable	71.22	8.86	0.466	658	9	658	-
TRITD7Av1G025830	Stable	71.76	9.33	0.516	2010	7	669	Endomembrane
TRITD2Bv1G237240	Stable	72.99	9.15	0.502	2031	13	676	Endomembrane
TRITD2Av1G273870	Stable	72.96	9.19	0.463	2031	11	676	Endomembrane
TRITD4Bv1G165010	Stable	75.69	8.88	0.404	2097	9	698	Endomembrane
TRITD5Av1G160650	Stable	35.2	8.81	0.316	954	5	317	Endomembrane
TRITD4Bv1G157680	Stable	71.31	9.01	0.504	1989	4	662	Endomembrane
TRITD4Av1G015910	Stable	70.99	9.19	0.511	1980	4	659	Organelle
TRITD5Bv1G135630	Stable	72.08	8.99	0.395	1971	3	656	membrane
TRITD3Bv1G170350	Stable	70.68	8.82	0.481	1947	9	648	Endomembrane
TRITD5Av1G144960	Stable	72.05	8.92	0.399	1971	7	656	-
TRITD3Av1G187350	Stable	70.66	8.85	0.483	1950	13	649	Endomembrane
TRITD5Bv1G102400	Stable	74.13	8.85	0.429	2058	7	685	Endomembrane
TRITD5Av1G124080	Stable	73.97	8.76	0.442	2055	8	684	Endomembrane
TRITD6Bv1G171900	Stable	70.22	9	0.541	1953	10	650	Endomembrane
TRITD6Av1G182150	Stable	70.01	8.85	0.536	1950	10	649	Endomembrane

## Data Availability

Not applicable.

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
