# Peer review of "Insights into the Sulfate Transporter Gene Family and Its Expression Patterns in Durum Wheat Seedlings under Salinity"

_genes, 2023, doi:10.3390/genes14020333_

Round 1

Reviewer 1 Report

1. General:

- The manuscript language is average; however, it needs several corrections, please check the entire manuscript for weak phrases and common structural mistakes.

- It is recommended to use the full scientific name for durum wheat by including the subspecies to avoid confusion with other subspecies. The full name should be: Triticum turgidum L. ssp. durum (Desf.). Please apply that to the entire manuscript including the title.

- Nomenclature of SULRTs is misleading. Based on previous articles for durum wheat, please change “TtSULRTs” to “TdAQPs” along the entire manuscript.  

2. Abstract:

- Line 11: replace “growth” with “plant”.

- Line 14: replace “Triticumturgidum” with “Triticum turgidum L. ssp. durum (Desf.)”.

- Line 15: change “in different” to “after different”.

- Line 16: change “exposed time series” to “exposure time period”.

- Line 17: this is well established for plant SULRTs, replace “five groups” to “the known plant five groups”.

- Line 20: change “LEU, VAL, and SER” to “L, V, and S”.

- Line 21: replace “site” with “sites”.

- Line 22: replace “target” with “be targeted”.

- Line 23: change “phytohormones which can more affect” to “plant bioregulators which potentially can affect”.

- Line 24: add “patterns” after “expression”.

- Lines 24-25: add “all” after “almost”.

- Line 26: remove “were”.

3. Introduction:

- Line 35: replace “plant performance” with “plant production”.

- Line 39: add “related” after “controlling”.

- Line 45: no need to abbreviate this region as it is mentioned just ONCE in the entire manuscript.

- Line 45: many SULRT designated articles classify them into 5 groups rather than 4. Please you need to cite them too!!!

- Line 63: replace “of S” with “of SULTR”.

- Line 69: replace “Triticumturgidum” with the abbreviated from “T. t. L. ssp. durum (Desf.)” or “T. turgidum L. ssp. durum (Desf.)” as it was written in full earlier in line 64.

- Line 70: replace “T. turgidum” with the correct abbreviated from “T. t. L. ssp. durum (Desf.)” or “T. turgidum L. ssp. durum (Desf.)”.

4. Materials and methods:

- Line 75: replace “T. turgidum” with the correct abbreviated from “T. t. L. ssp. durum (Desf.)” or “T. turgidum L. ssp. durum (Desf.)”.

- Line 77: access date should be mentioned for “Ensembl Plants database”.

- Line 78: replace “Triticumurartu, Triticumaestivum, Hordeumvulgare” with “Triticum urartu, Triticum aestivum, Hordeum vulgare”, with space between genus and species.

- Line 80: access date should be mentioned for “Pfam database”.

- Line 80: access date should be mentioned for “CDD”.

- Lines 82-83: access date should be mentioned for “ExpasyProtParam tool”.

- Line 84: access date should be mentioned for “BUSCA server”.

- Line 86: replace “T. turgidum” with the correct abbreviated from “T. t. L. ssp. durum (Desf.)”

- or “T. turgidum L. ssp. durum (Desf.)”.

- Line 85: replace “Evolution” with “Phylogenetics”. You did not investigate any evolutionary process!!!

- Line 101: access date should be mentioned for “Plant CARE tool”.

- Line 104: access date should be mentioned for “Phyre2”.

- Line 110: access date should be mentioned for “NetPhos 3-1 server”.

- Line 127: why “shoot parts” were ONLY collected. Do not you agree that you should have collected ROOTS too. According to different grouping of plant SULTRs (4 or 5 groups) group 3 or group 1, respectively, are expressed in ROOTS or as in group 2 in VASCULAR tissues!!!

5. Results:   

- Well presented, however, you need to check the language as indicated above.

- Lines 144-145: replace “Triticumturgidum” with the correct abbreviated from “T. t. L. ssp. durum (Desf.)” or “T. turgidum L. ssp. durum (Desf.)”.

- Figure 3: use one-letter code for amino acids along x-axis.

- Figure 5: adding percentage of related pie sections in “b” and “c” similar to “a” will be very helpful for readers.

- Lines 241-267: this is not enough at all! You need to mention the plant SULTR group to which these genes belong. AND their validated expression site in plant, e.g. roots and vascular tissues.

6. Discussion:

- Well presented, however, you need to check the language as indicated above.

- You need to discuss the plant SULTR group to which selected genes used in qRT-PCR belong. AND their validated expression site in plant, e.g. roots and vascular tissues.

- Previous similar studies NEED to be cited and discussed in the current study which include but not limited to:

Buchner, P., Parmar, S., Kriegel, A., Carpentier, M., & Hawkesford, M. J. (2010). The sulfate transporter family in wheat: tissue-specific gene expression in relation to nutrition. Molecular Plant3(2), 374-389.

Gallardo, K., Courty, P. E., Le Signor, C., Wipf, D., & Vernoud, V. (2014). Sulfate transporters in the plant’s response to drought and salinity: regulation and possible functions. Frontiers in plant science5, 580.

7. References:

- Please cite more recent articles, only 18 out of 54 were published in the last five years.

Reviewer 2 Report

The current paper titled with ‘Insights into the Sulfate Transporter…………under Salinity’ has focused on exploring Sulfate transporters (SULTRs) in Triticumt urgidum using insilico tools. Authors have also suggested that TtSULTR genes are involved in response to salinity in Triticumt urgidum. 

Following comments are for authors to further improve this manuscript. 

Line No. 14, 64, 69, 78 Check and rectify minor typos in botanical names provided. Please check entire manuscript for such errors. 

Is there any specific reason to use Actin7 gene as internal reference for real time PCR expression normalization? 

Authors are suggested to provide high resolution image and more detail in legends of figure 7 highlighting full name of closely interacting proteins with SULTRs. Moreover also provide confidence level of these interactions. 

Authors are suggested to improve conclusion part in light of their findings and their potential application in future. 

Line, 335-337, Please check sentence. Suggested to re-write for better understanding.

Reviewer 3 Report

The paper study is a very important issue for plant abiotic stress salinity for important economic crops. The paper is written in scientific format. The materials were sufficient. Introduction was clear. Discussion and conclusion were detailed. I recommend publish this paper.
